

# A Generalized approach to the operationalization of Software Quality Models

Clemente Izurieta[1,2,3], Derek Reimanis[3], Eric O'Donoghue[3], Kaveen Liyanage[4], A. Redempta Manzi Muneza[3], Bradley Whitaker[4] and Ann Marie Reinhold[2,3]

[1] Idaho National Laboratory, Idaho Falls, United States of America
[2] Pacific Northwest National Laboratory, Richland, United States of America
[3] Gianforte School of Computing, Montana State University, Bozeman, United States of America
[4] Electrical and Computer Engineering Department, Montana State University, Bozeman, MT, United States of America

## ABSTRACT

Comprehensive measures of quality are a research imperative, yet the development of software quality models is a wicked problem. Definitive solutions do not exist and quality is subjective at its most abstract. Definitional measures of quality are contingent on a domain, and even within a domain, the choice of representative characteristics to decompose quality is subjective. Thus, the operationalization of quality models brings even more challenges. A promising approach to quality modeling is the use of hierarchies to represent characteristics, where lower levels of the hierarchy represent concepts closer to real-world observations. Building upon prior hierarchical modeling approaches, we developed the Platform for Investigative software Quality Understanding and Evaluation (PIQUE). PIQUE surmounts several quality modeling challenges because it allows modelers to instantiate abstract hierarchical models in any domain by leveraging organizational tools tailored to their specific contexts. Here, we introduce PIQUE; exemplify its utility with two practical use cases; address challenges associated with parameterizing a PIQUE model; and describe algorithmic techniques that tackle normalization, aggregation, and interpolation of measurements.

Corresponding author
Clemente Izurieta,
clemente.izurieta@montana.edu

# INTRODUCTION

Software quality assurance (SQA) models are structured frameworks that assess the quality of software, typically by calculating an overall total quality index (TQI). The TQI indicates the degree to which measurements and observations of selected attributes meet the expectations of predefined criteria. Typically, these criteria combine high-level characteristics, such as maintainability, portability, scalability, security, efficiency, usability, reliability, and functionality. Often, these characteristics are delineated in standards such as the ISO/IEC 25010 (*ISO, 2011*).

While many factors affect an SQA model, the seminal step is selecting the high-level characteristics (sometimes specified by a standard) against which software will be measured

(*e.g.*, ISO/IEC 25010). Standards provide important theoretical definitions and descriptions of quality characteristics. However, standards do little to inform implementation. The theory described in the standards must be operationalized. Experimental comparisons of the SQA models Quamoco by *Wagner et al. (2015)*, QATCH by *Siavvas, Chatzidimitriou & Symeonidis (2017)*, and SQALE by *Letouzey & Coq (2010)* indicate a notable disparity between the quality assessments under these models. Causes for the disparity can range from the selection of tools, internal SQA mechanisms, and environment configurations. Thus, different operationalizations result in different findings.

Operationalizing a quality standard requires measuring key external quality characteristics using tools that can quantitatively collect selected metrics that serve as proxies for the quality characteristics. For example, *Cyclomatic Complexity* and *Code Density* can be used as metrics to address *Mainatainability*. Similarly, *Failure Rate* and *Mean Time Between Failures* (MTBF) can be used as proxy metrics for *Reliability*, and *Number of Vulnerabilities* can also be a metric for *Security*. Any candidate tool should be evaluated to ensure that the tool does not threaten the construct validity of the theoretical quality attribute the tool is intended to measure. In other words, can the metrics that a tool uses serve as proxies for the real world quality attributes they pretend to measure. While simple in concept, this evaluation is challenging in practice.

Tool selection is crucial and non-trivial. In SQA models, errors reported by tools informing a model create systematic errors that propagate, creating uncertainty that is challenging to constrain and quantify (*Izurieta et al., 2013*). This challenge is prevalent in SQA modeling because often different tools report different findings in the same software artifact, even when their intended purpose is the same as reported by *Reinhold et al. (2024)*. In addition, multiple versions of a single tool often report a wide range of findings in the same software artifact (*Reinhold et al., 2023*; *Reinhold et al., 2024*). Moreover, the number of findings reported by a tool is contingent on the configuration options of the tool and the host.

Once tools are selected, their findings must be combined to provide a comprehensive quality index. Combining results from multiple tools involves aggregating findings that may or may not measure the same quality characteristic. In addition, findings often must be aggregated despite reporting results on different scales (*e.g.*, ratio *vs.* ordinal) or ranges.

Aggregation typically requires defining various bottom-layer measures, such as lines of code (LoC) and number of classes (NC), and subsequently introducing a parent layer of normalized measures. Normalized measures are used to make software metrics comparable across projects by adjusting measurements based on the size of the code. This facilitates a way to compare various attributes of software quality, such as maintainability, in a normalized manner, making it easier to draw meaningful comparisons.

Aggregation methods can range from simple calculations of weighted averages to more intricate data transformations. Data transformations, particularly aggregation, threaten the internal and conclusion validity of an SQA due to the complexities and potential biases involved in combining multiple measurements into a single value. Ensuring accurate and representative edge weights is essential to maintain the validity of the model.

Internal validity refers to the extent to which a model accurately represents relationships between variables and conclusion validity refers to the ability to draw accurate conclusions from the results of the model. Data transformations, such as normalization and aggregation, can introduce biases if not handled properly. For example, if the edge weights used in aggregation are not representative of the actual importance of each measurement, the resulting aggregated value may not accurately reflect the true quality of the software and provide a wrong assessment of quality. This can lead to incorrect conclusions about the software's quality. Inappropriate edge weights produce systematic errors and uncertainties (*e.g.*, *Izurieta et al., 2013*) that propagate within a model. Thus, aggregation is a linchpin in constructing an SQA model.

Constructing a valid SQA model remains a wicked (https://en.wikipedia.org/wiki/Wicked_problem) challenge and measurability is one of the hardest open research problems (*The White House, 2024*). The quality index produced by an SQA model is a quantitative measure of interrelated qualitative constructs. Calculating a quality index is intrinsically complex and inherently subjective due to tool instability, changing requirements, and a lack of definitive measurement formulas for quality attributes. Also, calculating a quality index relies on selecting weights that assign importance to quality characteristics numerically.

Strategies for assigning numeric weights rely on manual inputs. Inputs are weighted using strategies that range in complexity from manual assignment to algorithms such as the analytical hierarchy process (AHP). For instance, Quamoco weights are created based on human-gathered importance orderings that are processed using the rank-order centroid method by *Barron & Barrett (1996)*. These weights are then applied uniformly across all characteristics within the quality model hierarchy. The evaluation moves up the hierarchy through a weighted summation process. Conversely, the QATCH quality model utilizes AHP (*Saaty, 2008*), a method that establishes a numeric ordering of importance *via* pairwise comparisons. Quality concepts, such as security, maintainability, or installability, serve as objects for comparison. An extension of this process is the fuzzy-AHP, allowing practitioners to input their uncertainty assessments for each pairwise comparison.

Uncertainty is an important factor in weighting strategies because parameterizing edge weights can introduce errors. Weighting requires some manual inputs, and the complexity increases with the growing number of characteristics in an SQA model. Thus, the ability of SQA models to support adaptive edge weighing is necessary for large models.

The complexity of large SQA models and the potential for errors to be introduced at multiple levels make effective visualization imperative. Because threats to validity can be introduced at each level, developers need the ability to visualize how their choices (*e.g.*, tools, edge weights) impact their models at each level and as a whole. However, to date, visualization tools for SQMs that our team has researched have been focused on the end-user rather than the developer. Thus, providing SQM developers with visualization tools to assist in assessing and debugging quality models (*e.g.*, normalization, aggregation, tool APIs) is a critical research objective.

Software quality modeling is a classic E-type system problem (*Lehman, 1980*). Despite the many challenges associated with software quality modeling, SQMs provide in-demand

information for government (*CISA, 2022*) and industry alike. Thus, software quality modeling remains an important objective for researchers and practitioners alike.

Given the challenges presented above, multiple opportunities for improvement exist in the design and development of quality frameworks. Although not comprehensive, we attempt to seize those opportunities to mitigate potentially significant threats to the internal and conclusion validity of models. To help us guide the research and development of a new quality framework, we asked the following research question (RQ):

## Research question

*Can we develop an extensible, flexible, and independent framework for operationalizing software quality across various domains, while enabling modelers to independently select their tools?*

We describe models created with our approach as Hierarchical Software Quality Assurance (HSQA) models. HSQA models are improvements to existing SQMs. HSQA models build on years of theory involved with hierarchical SQA. In addition, they incorporate multiple steps to reduce threats to validity and enhance the calibration and evaluation of software artifacts.

## Objectives and Contributions

Recognizing the limitations of current methodologies and the imperative to advance existing practices, this work's principal objective is to create a framework empowering software quality engineers to generate, validate, and operationalize quality models. The aim is to enhance the efficiency of efforts, facilitate experimentation, and foster collaborative opportunities. The contributions of this work encompass:

1. Platform for Investigative software Quality Understanding and Evaluation (PIQUE), an HSQA framework: Engineered to expedite the creation of experimental quality models that are easy to operationalize and emphasize improvements in aggregation techniques, Machine Learning (ML) weight selection, tool selection, and the overall ease of model generation.
2. Developer visualization tool: A tool tailored for developers, enhancing their ability to comprehend and engage with the intricacies of quality models.
3. Deployed HSQA models: Exemplars of operationalized HSQA models addressing diverse quality concerns, serving as practical demonstrations of the framework's capabilities.

We address these objectives as follows. In 'A Brief History of Software Quality Modelling' we provide a longitudinal description of related work beginning with early techniques in quality assurance modeling followed by hierarchical approaches alongside complementary tools. In 'A Generalized Approach', we explore the PIQUE framework and describe its internal structure, metadata model, the software components that make the core framework, the tool selection process, and the generation, execution, and visualization of PIQUE models. 'Internal Framework Mechanics' focuses on the internal mechanisms that PIQUE uses to calculate a quality score, specifically, normalization, score mapping, aggregation, and ML weight selection. In 'Use Cases' we exemplify the PIQUE framework

through two use cases. Finally, 'Threats to Validity and Conclusions and Summary' describe the threats to validity and conclusion respectively.

# A BRIEF HISTORY OF SOFTWARE QUALITY MODELLING

Research in SQA can be dated back to the 1970s. In 1976 *Boehm, Brown & Lipow (1976)* introduced the first quality model highlighting three top-level software qualities: utility, maintainability, and portability. Following this, in 1977, *McCall (1977)* proposed a more compact hierarchical model that only had two levels, and subsequently, in 1978 *Boehm et al. (1978)* expanded on their seminal work. Both models presented similar decompositions of software quality by employing abstract characteristics which were then decomposed into more concrete sub-characteristics. Although the modern ISO/IEC 25K standard is divided into five separate divisions, many of its categories (*i.e.,* Quality In Use and Software Product Quality) can be directly linked to the principal concepts of earlier models.

In the 1980s we see additional models that expand on early work. *Grady & Caswell (1987)* also propose a hierarchical decomposition of categories. This research led some software companies to adopt variations of these early models. The Functionality, Usability, Reliability, Performance, and Scalability (*i.e.,* the FURPS) model by *Grady (1992)* for example, was used extensively by companies like Hewlett Packard Co. during their software lifecycle and especially during the final quality assessment of their software (*e.g.*, HP-UX Operating System releases). *Eeles (2005)* expanded upon this model by introducing additional sub-characteristics, labeling it FURPS+, and integrating it into the IBM Unified Rational Process led by *Kruchten (2000)*. In the FURPS models, each characteristic of quality required test engineers to identify suitable metrics to satisfy release criteria. These models are perceived as early operationalized frameworks adopted by industry.

Early research in software quality models and feedback from their early commercial use also revealed the need to make tradeoffs and balance the influence of sub-characteristics to help adjust scores to meet contextual constraints. The breakdown of quality characteristics, along with acknowledgment of the weighted influence of these categories, led to the formation of the initial ISO standards and hierarchical evaluation techniques for quality, which were also impartial to specific products.

## Modern hierarchical models

The 1990s mark a transition for software quality models. This decade sees a shift to vendor-neutral standards. In 1991, the ISO/IEC 9126 (*ISO-IEC, 1991*) surfaced as the first theoretical hierarchical model with a limited number of high-level characteristics. This model, however, lacked important quality aspects and evolved into the ISO/IEC 25010 SQuaRE model (*ISO, 2011*), which kept the hierarchical approach but expanded upon its predecessor by adding the security and compatibility characteristics.

Despite improvements, these models only focus on the top half of the hierarchy (*i.e.,* external quality characteristics), leaving out necessary details to make them useful. *Van Zeist & Hendriks (1996)* and *Samoladas et al. (2008)* both presented extensions to the ISO/IEC 9126. *Van Zeist & Hendriks (1996)* collaborated with six established companies

to produce the QUINT (Quality in Information Technology) framework. The first version resulted in a handbook (*Punter, van Solingen & Trienekens, 1992*) describing a software quality model and guidelines on how to use it. The second version of QUINT extended the framework by referencing the Extended ISO 9126 model. *Samoladas et al. (2008)* quality model focused on Open Source Systems (OSS) and extended some characteristics based on the ISO/IEC 9126. Although other quality models had tackled OSS systems, they required significant effort to set up. In this model, the authors required limited user interaction once the profile of the quality model assessment was established.

One of the earlier operational hierarchical models focused on object-oriented systems. The Quality Model for Object Oriented Design (QMOOD) software quality model was proposed by *Bansiya & Davis (2002)* and was based on *Dromey (1995)* which presented a quality model that used a hierarchy based on structural concepts of a system (class, function, object) instead of quality concepts such as maintainability, reliability, and usability. QMOOD is based on six quality characteristics that tie the quality of code to the quality of the design. The model is still used today and is effective in several open-source and e-commerce studies. However, operational constraints hinder the metrics' effectiveness across software updates and releases. The advent of Agile, DevOps, and Continuous Integration/Continuous Development (CI/CD) characterize new development approaches marked by frequent software changes. The instability of these environments with integrated stakeholder feedback, significantly diminishes the efficacy of the QMOOD metric suite.

In 2003, *Franch & Carvallo (2003)* presented a variety of metrics in an attempt to use quality models to assess the quality of software package selection. Most methodologies proposed for choosing software packages compared user requirements with the packages' capabilities rather than focusing on quality requirements. The authors attributed this to the lack of package descriptions and their corresponding quality requirements in specific domains. They proposed a six-step method to operationalize the ISO/IEC 9126 model.

The SIG maintainability model (*Heitlager, Kuipers & Visser, 2007*) was introduced as a distinct approach that only presents a maintainability model. This model assesses sub-characteristics related to maintainability by contrasting them with five source code properties: volume, complexity per unit, duplication, unit size, and unit testing. This approach coincides with efforts to focus on parsimonious models of characteristics that can be operationalized and that are well understood, or that resonate with software developers. The focus on maintainability coincides with the efforts of the technical debt community (*e.g.*, *Izurieta et al. (2018)*; *Izurieta & Prouty (2019)*) where there is now consensus to address maintainability issues earlier rather than later. "Technical Debt presents an actual or contingent liability whose impact is limited to internal system qualities, primarily maintainability and evolvability" (*Avgeriou et al., 2016*).

Introduced in 2009, the SQUALE model and framework (*Mordal-Manet et al., 2009*) is founded on the ISO/IEC 9126 quality model, emphasizing practical applicability in industrial settings. SQUALE enhances ISO/IEC 9126 by incorporating a more detailed intermediate layer through a concept known as practices. While it furnishes additional metrics for in-depth assessment, the framework includes valuable tool support and

visualization features. However, the model lacks modularization and doesn't establish a robust link between low-quality scores and specific low-level code issues.

The SQALE model by *Letouzey & Coq (2010)*, introduced in 2010, adopts a distinct approach to quality modeling by associating quality with technical debt values determined by remediation costs. It organizes the model into hierarchical layers, ranking them using calculated values of testability, reliability, changeability, efficiency, maintainability, and reusability. Notably, SonarQube (https://www.sonarsource.com), a widely used framework and continuous integration quality monitoring service, employs SQALE as its foundational quality model for assessments.

The PIQUE framework presented in this study is preceded and influenced by multiple publications on quality models, namely, Quamoco by *Deissenboeck et al. (2011)*, *Wagner et al. (2012)*, *Wagner et al. (2015)*, *Izurieta, Griffith & Huvaere (2017)*, and the Quality Assessment Tool CHain (QATCH) by *Siavvas, Chatzidimitriou & Symeonidis (2017)*. These models bridge the gap between the low-level layers representing directly measurable metrics, and the higher-level layers representing more abstract theoretical quality aspects. The authors of Quamoco explicitly stated, "Our aim was to develop and validate operationalized quality models for software together with a quality assessment method and tool support to provide the missing connections between generic descriptions of software quality characteristics and specific software analysis and measurement approaches."

*Kitchenham et al. (1997)* recognized the need for a meta-model in their early SQUID approach to describe the complexity of potential instances of models. Quamoco also includes a meta-model that is generic and extendable, modularized, and integrated with benchmarking data. Developed in 2017, QATCH is a quality modeling project that, unlike the Quamoco project, doesn't adhere to a meta-model description and lacks extensive validation efforts. However, QATCH stands out by concentrating on an automated approach to generate quality models that are attuned to and responsive to the subjectivity of stakeholders. In contrast to the Quamoco model, the number of layers in QATCH is restricted to three. The authors argue that the benefits of comprehensibility, explainability, and ease of extension their model brings outweigh the loss of granularity brought by large, complex models. The QATCH model also introduces the Analytical Hierarchy Process (AHP) (*Saaty, 2008*) that uses pairwise comparisons between characteristics to derive a numeric ordering of importance to high-level aspects of quality. AHP can be used as a rudimentary way to elicit the model's higher-level weights. The obvious drawback is that as the number of characteristics increases, the pairwise comparisons square in size.

A current effort in quality modeling is the contribution made by *Bass, Clements & Kazman (2021)* from the Software Engineering Institute (SEI). They attempted to provide an alternative to the ISO/IEC 25010 standard by significantly reducing the high-level characteristics and sub-characteristics from approximately 40 to 10. Reservations remain regarding the content validity of this model.

## Measurement and Tools

Numerous tools can inform HSQA models by providing critical data that help assess many aspects of software quality. These tools can be broadly categorized into several

types based on their functions and the aspects of quality they address, however, PIQUE focuses on Static Analysis Tools (SATs) that capture metrics of source code or binaries. Other categories such as dynamic analysis, code coverage, and continuous integration tools are not specifically addressed. The comparison and contrast of these tools is beyond the scope of this paper; however, this section is meant to help communicate the expanse of available tools, in particular, the tools that the authors have been exposed to. These tools are available for conducting static analysis of source code, binaries, Software Bill of Materials (SBOM) components, cloud services, and microservices packages. Examples include tools for assessing coding styles (*e.g.*, Checkstyle (https://checkstyle.sourceforge.io/)) and identifying source code warnings and bugs (*e.g.*, Lint (https://man.freebsd.org/cgi/man.cgi?query=lint&manpath=FreeBSD+11.2-RELEASE), PC-Lint (https://pclintplus.com/), FindBugs (https://findbugs.sourceforge.net/), PMD (https://pmd.github.io/), Roslynator (https://github.com/dotnet/roslynator), Security Code Scan (https://security-code-scan.github.io/), SonarQube, and Insider (https://github.com/insidersec/insider). Further, specialized tools focus on binary analysis, with the aim to detect potential security threats, weaknesses, and vulnerabilities in a compiled program. Examples include CVE Binary Tool (https://github.com/intel/cve-bin-tool) ("cve-bin-tool" developed by Intel) and cwe-checker (https://github.com/fkie-cad/cwe_checker) (developed by the German research organization Fraunhofer FKIE). Tools to help assess SBOM analysis include Grype (https://github.com/anchore/grype) and Trivy (https://github.com/aquasecurity/trivy).

# A GENERALIZED APPROACH

The PIQUE framework represents a progression from prior quality models, involving multiple iterations and enhancements across various components of this complex software suite. *Rice (2021)* adopted techniques from earlier quality models into the initial versions of PIQUE. The framework has undergone many refinements from these earlier techniques, resulting in improvements to the software framework.

We describe the details of PIQUE's model structure in 'Model Structure', the process of selecting tools that inform the model in 'Tool selection', and the iterative process to generate an operationalized PIQUE model in 'Model Generation'. The steps to generate a PIQUE model are as follows:

1. Determine model structure
   (a) Select high level quality characteristics of the model
   (b) Determine the number of layers needed to represent subcharacteristics of each high level characteristic
   (c) Determine the relative importance of characteristics and subcharacteristics by assigning weights
2. Select the SATs that will inform the operationalization of PIQUE
3. Model generation
   (a) Determine PIQUE model interfaces to implement
   (b) Determine behaviors for aggregation, normalization, and utility function scoring

(c)  Determine a benchmarking corpus (if available) to compare against and run PIQUE in benchmarking mode

(d)  Instantiate PIQUE and run it in assessment mode against a target

(e)  Iterate and refine the model

Figs. 1 and 2, depict the structure and the operationalization of PIQUE respectively. The methodology used to develop this generalized approach encapsulates ideas from prior art and multiple iterations of improvements to the internal mechanisms (see 'Internal Framework Mechanics') of PIQUE informed by multiple deployments of this technology. Operationalized versions of PIQUE are presented in 'Use Cases', illustrating its application *via* exemplary use cases.

## Model Structure

PIQUE (https://github.com/MSUSEL/msusel-pique) features the breakdown of quality-related characteristics into a hierarchical tree-based data structure. In Fig. 1 we depict a generalized view of the structure of PIQUE, and in Fig. 2 we depict the general approach to generating and operationalizing a quality assurance model based on the PIQUE technology. Because PIQUE is tool agnostic, we can leverage multiple tools and interface them with PIQUE, thus allowing practitioners a choice in using their preferred tools. PIQUE models are hierarchical and provide layered and holistic overviews of assets. The technology leverages existing static analysis methods as inputs and scores the quality and security of software artifacts. Scores are provided at multiple levels in the hierarchy—with audiences that range from developers to project managers to C-suite executives. Troubleshooting at lower levels allows developers to understand scoring techniques and calibrate tool inputs to the model, whereas the higher levels of the PIQUE model provide summaries (albeit, with a loss of accuracy) that target management and security operations.

PIQUE is a collection of library functions and runner entry points designed to support experimental software quality analysis from a programming language and component-agnostic perspective. To remain agnostic, the PIQUE framework is structured as a library that provides the abstractions, interfaces, and algorithms necessary for quality assessment but leaves the task of defining specific analysis operations to dependent projects. To improve adoption, this framework provides default classes for each quality assessment component, and thus allows the platform to be used "out of the box." For experienced practitioners with quality assessment approaches, the platform allows each component to be overridden with experimental approaches.

### Software components

The PIQUE framework has five software components that work together to achieve quality benchmarking and assessment: Runner, Analysis, Calibration, Model, and Evaluation. The PIQUE framework is viewed as a library, and a model developer must implement relevant classes/interfaces from the library to build a functional PIQUE model.

**Runner:** The Runner component provides the abstractions necessary to automate three software components necessary for quality assessment in PIQUE: (1) Deriving a quality model, (2) benchmarking the quality model based on similar projects, and (3) using that model to assess the quality of a system.

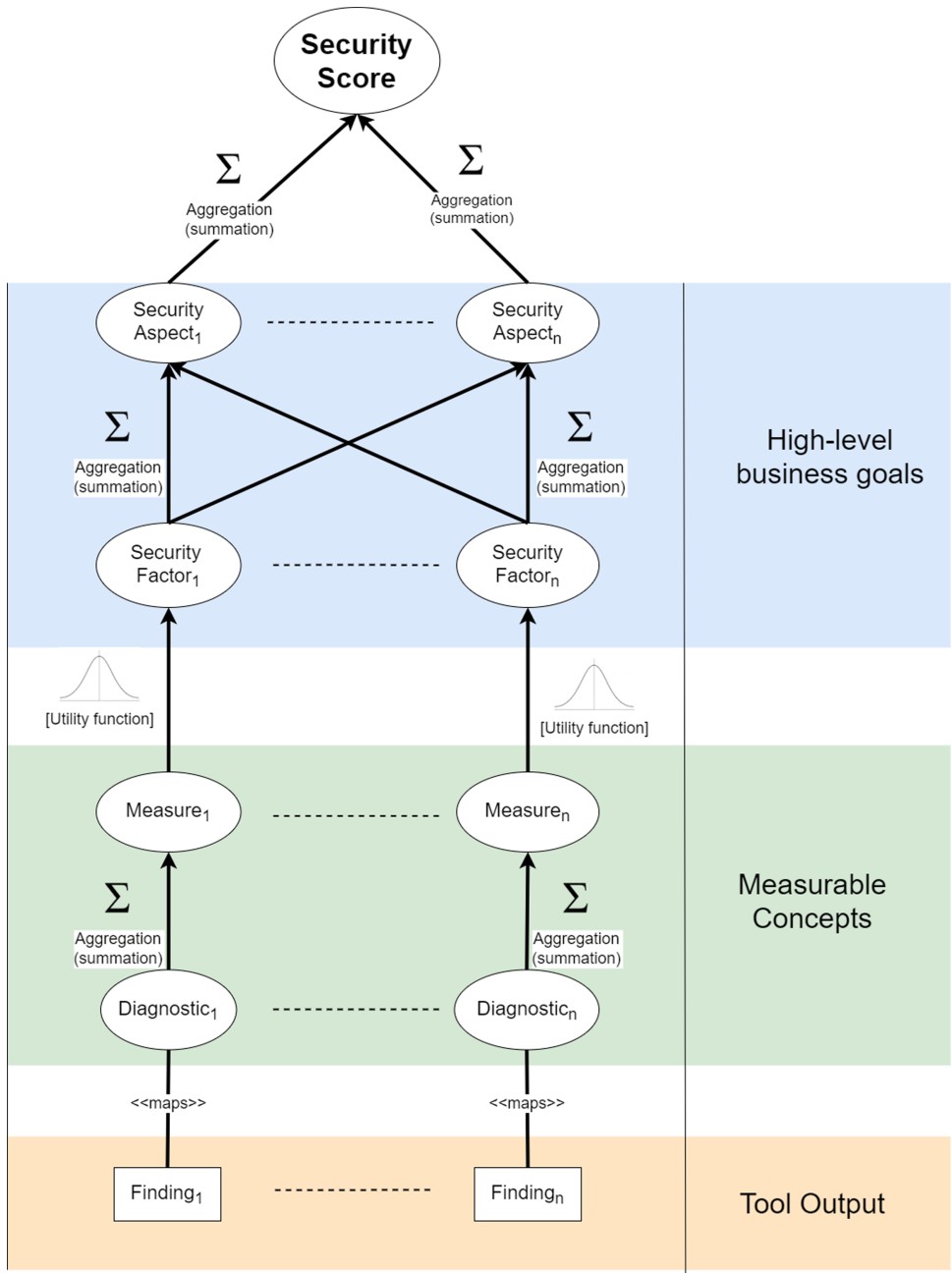

**Figure 1** **High-level depiction of the structure of a PIQUE model.** In this figure, we use PIQUE to assess the quality associated with security aspects only. The tree structure is subdivided into high-level business goals that typically represent abstract concepts. Further down the hierarchy, we depict measurable concepts, and in the lowest level of the tree, we depict the tool layer. Findings from runnable tools propagate up the tree according to various aggregation, normalization, and utility function techniques that are either preconfigured or extended by a model developer.

**Analysis:** The Analysis component provides the abstractions necessary for specific PIQUE extensions to instantiate analysis tools connected to the model. For every analysis

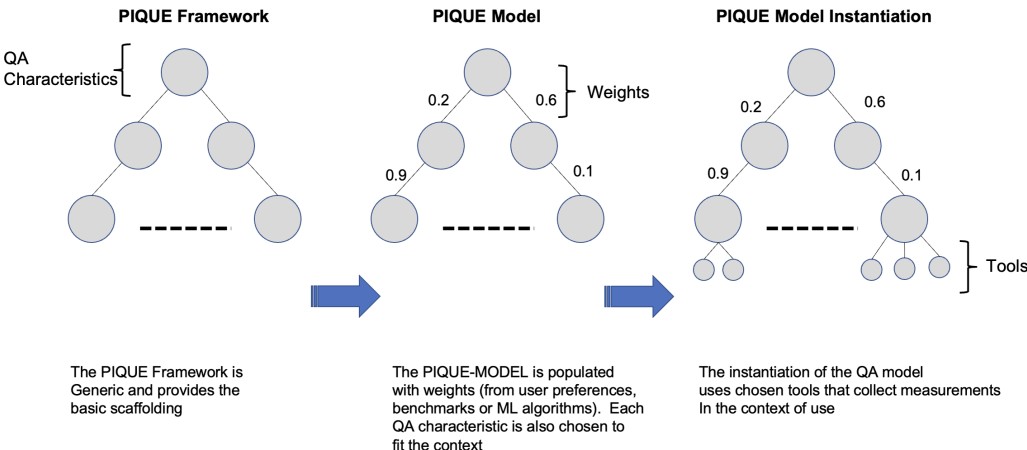

**Figure 2   Process for generating an operational PIQUE model.** From left to right, a model developer begins the process by using a generic model that is customized with QA attributes commensurate with the target domain (*e.g.*, MS STRIDE), then the importance of the contributions of various attributes at lower levels is captured through a weighing approach. Finally, tools available in the domain are connected to the model to achieve full operationalization.

tool in the PIQUE model, these abstractions in the Analysis component include function definitions to (1) Programmatically run the analysis tool and (2) Parse the output from the analysis tool into Finding or Diagnostic nodes.

**Calibration:** The Calibration component provides the abstractions necessary to allow modelers to prioritize particular characteristics over others, including the implementation of domain-pertinent information. The Calibration component consists of two modules, (1) Weighter, which provides classes that implement the logic to assign edge weights within the model, and (2) benchmarker, which provides classes to perform benchmarking, a process that involves establishing the distribution of every analysis tool result. Default classes are provided for each module within the Calibration component, yet model accuracy may improve by extending to the classes of this component.

**Model:** PIQUE is, at its core, an implementation of the family-tree data structure, a generalization of the tree data structure in which any node can have more than one parent. The Model component provides the abstract and concrete classes necessary for the structure of the model, namely classes for each of the model layers discussed in Table 1. Additionally, the Model component provides concrete classes with the functionality to serialize and deserialize the model. Generally, classes from the Model component are not extended because they are fundamental to the vision of PIQUE.

**Evaluation:** The Evaluation component provides the necessary functionality for quality model evaluation and assessment. It provides the algorithms and strategies used for model evaluation (*i.e.,* the calculation of the TQI); specifically the (1) Normalizer, which provides classes that normalize findings based on system meta-data, (2) Evaluators, which provide strategies required to aggregate values between layers, and (3), Utility Functions, which provide the algorithms to perform interpolation on the distributions found from findings

**Table 1  Definition of terms associated with each level of a PIQUE tree.** The table begins with definitions of the highest-level nodes and ends with nodes directly connected to tools. PIQUE builds on existing literature from *Wagner et al. (2012)* and adopts prior terminology.

| Node | Definition | Resource |
|---|---|---|
| Quality aspects | High-level factors that express abstract quality goals that cannot be measured directly. | *Wagner et al., 2012*, 101–123 |
| Product factors | Nodes that can decompose into directly measurable concepts, generally attributes of the parts of a product or subfactors of quality aspects that further concretize a system. | *Wagner et al., 2012*, 101–123 |
| Measures | Concrete definitions of product factor values. A measure holds the knowledge of its relevant benchmarked utility functions and contains the evaluation information needed to calculate its value from incoming diagnostics. A measure provides a 'score' of a particular analysis result, scoring the analysis result from a system under evaluation to every similar analysis result from a collection of similar systems (also referred to as the benchmark repository). | *Wagner et al., 2012*, 101–123 |
| Diagnostic | A representation of the parts needed for a measure to evaluate. A diagnostic must evaluate directly from the results of its connected tool's output. | *Rice* (*2021*, 48–50) |
| Finding | Data object representation of a "hit" from its associated analysis tool. A finding is only instantiated after its associated tool has run an audit on the system under evaluation. | *Rice* (*2021*, 48–50) |

on the benchmark repository. Default implementations of each are provided by default, yet model accuracy may improve by finessing or extending these classes.

### PIQUE metamodel

The design of the PIQUE framework metamodel is shown in Fig. 3 and comes from reviewing the strengths and weaknesses of previous modeling approaches, and is heavily influenced by the QATCH and Quamoco hierarchical models. The former was designed with simplicity in mind by keeping the number of layers constrained to three (*i.e.,* characteristics, properties, and measures layers), while the latter allowed for arbitrarily deep hierarchies consisting of characteristics, aspects, factors, and measures, where factors could also be subdivided into more sub-factors. In PIQUE we balanced these approaches and subdivided the hierarchy (from top to bottom) with aspects, factors, measures, diagnostics, and findings, where the terms are defined in Table 1.

These terms provide descriptions of concepts that serve as an initial starting point for hierarchical model design, yet are flexible enough to allow for interchangeability based on the scope of the system under analysis. In our observations, we've discovered that practical scenarios, such as specific stakeholder demands and situations necessitating systems analysis, often determine whether the node embodies a more abstract or concrete meaning.

### Tool selection

Tool selection requires careful consideration of the underlying model. PIQUE features the breakdown of quality-related characteristics into a hierarchical tree. The characteristics and sub-characteristics comprising the PIQUE model are selections made by domain experts. For example, a practitioner wishing to evaluate the quality of source code in a given system

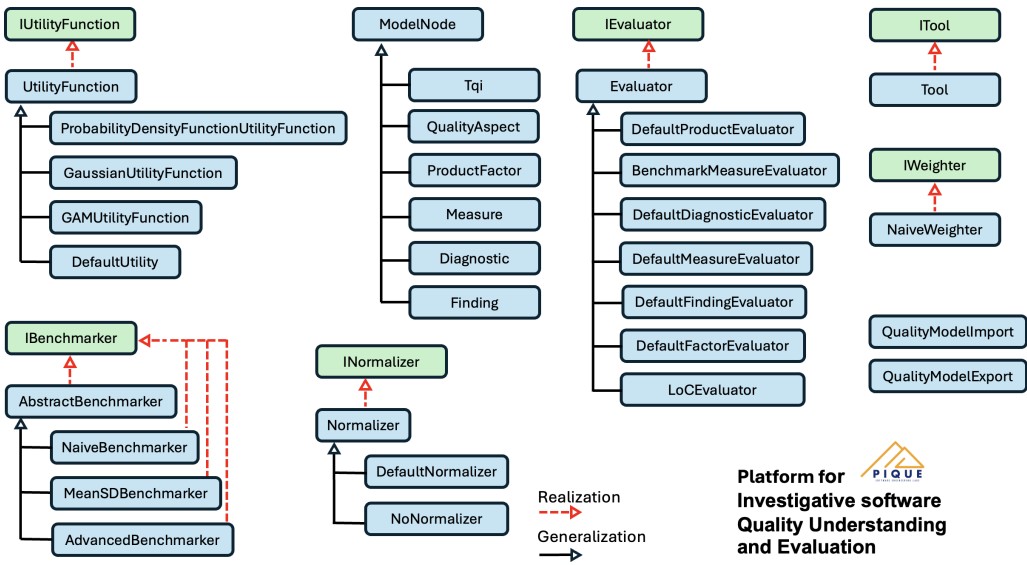

**Figure 3** **The PIQUE framework metamodel.** The structural model is drawn using the Unified Modeling Language (UML) where red dashed arrows represent interface realizations and black solid arrows represent class inheritance.

will likely select an ISO model (*e.g.*, ISO 25K) as the top-level characteristics, whereas a practitioner focusing on security may select Microsoft's STRIDE model as the top-level characteristics. Further, quality is subjective, and the construction of a hierarchical model can be tailored to adjust to new notions and interpretations of quality that potentially combine underlying models, reference documents, or characteristics from diverse sources. For example, the quality of an SBOM artifact is of high importance to help assess the security of the supply chain and could combine notions of adherence to SBOM standards as well as the quality of the software represented and its dependencies.

The top-level characteristics help modelers with the selection of complementary tools. The tools selected should ideally meet the representation condition (*Fenton & Pfleeger, 1996*) of the measured theoretical characteristics (and sub-characteristics). Poor representation increases the construct validity of these models. As shown in Fig. 2, tools are connected to the model during PIQUE model instantiation. Importantly, we emphasize the ability of an organization to leverage existing tool investments in their organizations. By leveraging and combining tools we help with the generation of a holistic review of quality in context. Our experience selecting tools during the development of various PIQUE models is shown in Table 2. We also provide information describing the underlying model (*i.e.,* characteristics) used to compare operational results against. Measurements obtained from tools are diverse (*i.e.*, different ranges, scales, etc.) and the processes of aggregation and normalization are explained in detail in 'Internal Framework Mechanics'.

## Model Generation

Generating a PIQUE model involves several steps including model design, model implementation, benchmarking, and finally model assessment. The greater process is

**Table 2  Metadata associated with various models was used to build PIQUE models.** The first column shows the metadata. The underlying model describes the reference(s) used to select top-level characteristics in a PIQUE model. The High-Level Characteristics are the selected Quality Aspects in the PIQUE model, and the Tools represent the best available tool selected as a proxy to measure the characteristics.

| PIQUE | Software quality | Software security | Binaries | Programmable logic controllers | Cloud | SBOM security |
|---|---|---|---|---|---|---|
| Underlying Model | ISO 25010 | STRIDE | STRIDE CWE-699 | Top 20 Secure PLC Programming Practices (https://github.com/VirusTotal/yara) Valentine Taxonomy (*Valentine, 2013*) PLCOpen Guidelines on Software Quality Metrics (https://github.com/VirusTotal/yara) | ISO 25010 CWE-1000 | ISO 25010 security section STRIDE CWE-699 |
| High-Level Characteristics | Functionality, Performance, Compatibility, Usability, Reliability, Security, Maintainability, Portability | Spoofing, Tampering, Repudiation, Information disclosure, Denial of service, and Escalation of privileges | Spoofing, Tampering, Repudiation, Information disclosure, Denial of service, and Escalation of privileges | Reusability, Testability, Efficiency, Maintainability, Reliability, Rule-based Issues | Functionality, Performance, Compatibility, Usability, Reliability, Security, Maintainability, Portability | Confidentiality, Integrity, Non-repudiation, Authenticity, Accountability, Availability, Authorization |
| Tools | Roslynator | Security Code Scan Insider | cwe_checker CVE-bin-tool Yara (https://github.com/VirusTotal/yara) with rule repository 'yara-rules' (https://github.com/VirusTotal/yara) | CODESYS (https://github.com/VirusTotal/yara) | Grype Trivy | Grype Trivy |

analogous to an iterative lifecycle from a Software Development Life Cycle (SDLC), with opportunities to cycle on previous steps based on stakeholder feedback. We also enhance the process with a visualization tool described in the last subsection. Visualization tools are lacking and are needed by modelers.

### Model Design

Model generation begins with model design. Model design is analogous to the requirements stage from the SDLC, and is a soft process (rather than a technical process) that involves meetings and iteration with project stakeholders. We have found that applying a rigorous process in this stage results in unresponsive stakeholders and that a level of informality and camaraderie expedites the process. Artifacts in this stage include meeting minutes, emails, and drawings or whiteboarding pictures. We usually initiate this procedure by pinpointing two types of stakeholders: (1) those with a business interest and (2) those with technical expertise, and then investigate the overarching purpose behind employing the model. For instance, we inquire with stakeholders whether they seek the model to prioritize security. This allows us to ensure the final model will apply to end users. We have found that stakeholders prefer to provide a general design direction by selecting high-level quality

characteristics (featured in the quality aspects and occasionally the product factors layers of PIQUE) and leaving specific design details to model developers. Largely, this means model developers perform research to identify the state-of-the-art measurements pertinent to the technology associated with the project under analysis, with the caveat that tooling exists to capture such measurements. This process is iterative, so modelers are keen to open discussion and re-design if stakeholder views change.

### Model implementation

With an initial model design complete, model implementation can begin. Model implementation is a multi-faceted process that begins with the creation of a model definition file. The model definition file is a flat JSON file that captures the structure of the model. Specifically, this file contains attributes for every node in the model, including the node name, child nodes, utility functions, and placeholders for node values. Node values are calculated during the benchmarking and assessment phases, described in 'Model Benchmarking and Model Assessment'.

Beyond the creation of the model definition JSON file, model implementation generally involves implementing and extending the PIQUE framework, written in the Java language version 1.8. The PIQUE framework can be implemented with minimal code additions, specifically the development of Driver classes that provide the runtime system with hooks to run the model and the development of Tool Wrapper classes that provide the functionality to (1) programmatically run tooling and (2) parse the results into the model, every separate PIQUE model we have developed requires certain customizations. For example, several PIQUE models utilize vulnerability information in the form of CVEs, which have severity information embedded, thus PIQUE framework extensions are required to incorporate the severity information into the model. The PIQUE framework has been designed specifically with these extensions in mind, expediting the model implementation process.

Once the model is deployed, it's prepared for benchmarking, which requires executing the code from start to finish.

### Model benchmarking

PIQUE models are rooted in comparison. Benchmarking is the process by which the quantitative basis for comparison is created. This process involves executing the PIQUE model on a number of projects (referred to as the benchmark repository) that are similar in context and quality concerns to the ultimate project under analysis. Modelers generate distributions for each finding reported by the static analysis tools so that interpolations on the distribution can be performed during the model assessment process. Once the PIQUE model has been implemented, benchmarking is initiated by executing the model with a program flag. When executed with the benchmark flag, PIQUE first parses the model definition to generate a persistent model in memory, and then, executes every static analysis tool on every project in the benchmark repository. As tools complete their runs, output from the tool is parsed into a numerical representation that aligns with the selection of a utility function; for example, if the selected utility function is a simple linear interpolation, the numerical representation consists of the minimum and maximum counts of each finding, the values of which are used to perform linear interpolation in the model

assessment stage. Different utility functions have different numerical representations. After every tool has been executed on every project in the benchmark repository, the model is serialized to a JSON file, which can be used as input into the PIQUE model during the model assessment. This JSON file is the output artifact from the model benchmarking process.

A key focus in this process is the challenge of choosing projects that accurately represent the final project under analysis. The problem can be divided into two subproblems, (1) identification and (2) retrieval of projects. To address the identification of projects, we identify qualitative properties from the project under analysis, such as system domain, and scour code repositories for projects that match qualitative properties. To address the retrieval of projects, we rely on manual downloading and storing of project files because of the difficulty of programmatically downloading many project files. We recognize that this process introduces selection bias and the impacts of this are discussed in the threats to validity section, as construct and external validity.

### Model assessment

After model benchmarking, modelers perform the assessment. The model assessment process involves running the PIQUE model with a flag that specifies to PIQUE that it should run the evaluator. The underlying code and processes are the same as with model benchmarking, with the exception that the finding distributions identified in the benchmarking process, are interpolated based on the findings reported from the static analysis tools when targeting the project under analysis. The output values from interpolation are aggregated to higher-level nodes in the model tree through different aggregation techniques, specified by the evaluator and the weighter nodes. Evaluator nodes handle the algorithms that compute the value of the node. The default algorithm calculates the average of the value of all child nodes weighted by edge values, and the weighter nodes handle the strategy by which edge weights are assigned. The output artifact from the model assessment process is a JSON file with values at every node, including a value for the root node in the tree which captures the TQI.

### Visualization

Although many point tools do offer graphical user interfaces, our goal is to improve visualization in frameworks to capture holistic scores (*i.e.,* a score based on the aggregation of potentially many tool scores). One visualization tool we found that also aims to address the needs of managers as well as model developers is exemplified in work by *Martinez-Fernandez et al. (2019)*. The authors provide a strategic and a raw data dashboard to visualize information. We have also developed dashboard technology that aims to address the concerns of C-suite level managers by focusing on the high-level characteristics of a PIQUE model, as well as developer/modeler technology that allows a practitioner to focus on the details associated with the nuances of scores calculated at the lowest levels of the PIQUE models. The PIQUE dashboard is informed with JSON files that are output from the PIQUE framework.

At the higher layers of the model, the scores represent the outlook of a system that is of interest to program and project managers. This is shown in Fig. 4A. The lower you move in

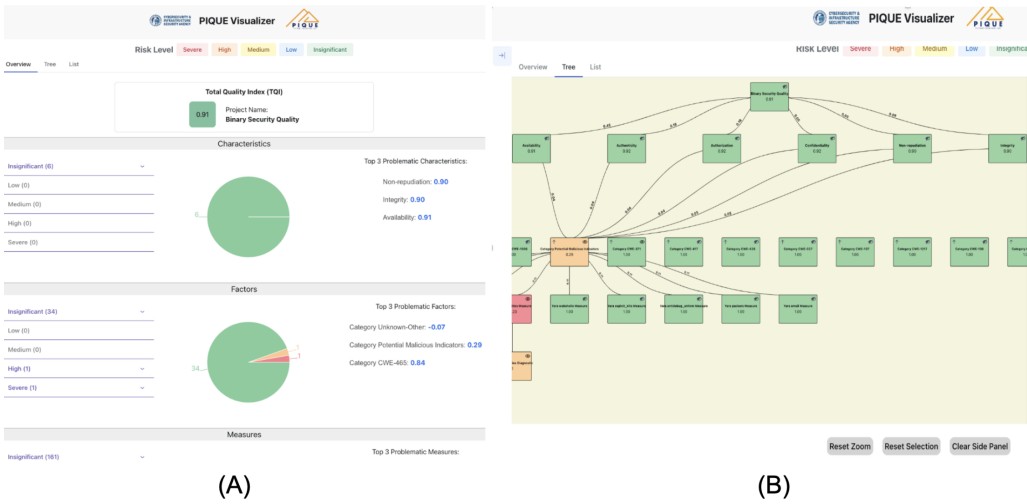

**Figure 4** (A) **Visualization dashboard of high-level characteristics of a PIQUE model.** Users can access pie charts and aggregated scores at the Aspect levels (B) Visualization of lower layers of the PIQUE model. At the lowest level, developers and modelers can review actual scores produced by tools and the weights associated with the edges connecting hierarchical levels of the tree. This figure is courtesy of the Software Engineering Laboratory at Washington State University.

the layers of the model, the closer you get to the sub-characteristics of interest to developers and modelers. This is shown in Fig. 4B.

# INTERNAL FRAMEWORK MECHANICS

Four critical mechanisms can have profound effects on quality scores. All mechanisms impact the accuracy of the TQI and mitigating the variability of the sources that feed and inform a PIQUE model. Specifically, we discuss how normalization, score mapping, aggregation, and the use of ML can improve the accuracy of the TQI.

## Normalization

Normalizing values associated with metrics and finding counts allows modelers to compare and combine scores from disparate sources. In PIQUE, the DefaultNormalizer software component is provided with the framework and is the default algorithm used; it divides the value of a node by the LOC in the target project. However, the diversity of the software artifacts makes normalization a challenge.

When normalizing projects across multiple languages, it is more likely that modelers will use a metric suite like Halstead's complexity metrics described by *Remoortere (1979)*. Modelers assessing binary file quality may first produce a high-level representation of the program that includes reconstructed control flow elements suitable for human consumption or machine-based inspection. Metrics from the control flow can then be used to normalize measures. Finally, if modelers are comparing SBOM files, they may focus on normalizing based on the number of third-party dependencies associated with each SBOM.

## Score Mapping

PIQUE uses utility functions to map values or scores between layers to an appropriate range. The utility function provides normalization at the most atomic level in a PIQUE model. PIQUE has been designed so that every node in PIQUE has a unique utility function, one of which can be the identity function (https://en.wikipedia.org/wiki/Identity_function). In general, each utility function measures some combination of findings for a specific software project and outputs a value based on how it compares to other software projects in the benchmark repository. Our original utility-function algorithms are based on the QATCH operational framework from *Siavvas, Chatzidimitriou & Symeonidis (2017)*. Currently, multiple utility functions are available in PIQUE, but the default is a variation of the Linear Interpolation utility function. This utility function ingests the minimum and maximum values found within the benchmark repository for each measure and outputs a score based on interpolating the value for the project under analysis.

In our early research, we extended the work of *Siavvas, Chatzidimitriou & Symeonidis (2017)* by offering a *NaiveBenchmarker* and a Gaussian-function-based Benchmarker, named *BinaryBenchmarker*. The *NaiveBenchmarker* calculates the lowest and highest Diagnostic value, wherein each Diagnostic value is generally a sum of Findings. This function generates values that are used as thresholds and later ingested in the utility function. In contrast, the *BinaryBenchmarker* calculates the mean plus or minus the standard deviation of each Measure to produce the threshold values. Subsequent versions of PIQUE introduce two additional utility functions: the *GAMUtilityFunction*, and the *GaussianUtilityFunction*, however, these functions are deprecated. PIQUE's design is flexible, allowing for any Benchmarker to be used for calculating thresholds and providing the framework to define custom Benchmarking functions based on the nature of the data.

In our current framework, we use a scaling approach. This scaling function compares each artifact against the benchmarks using density-based scoring. This method is termed the *ProbabilityDensityFunctionUtilityFunction*, herein referred to as a *PDFUtilityFunction*. This function is depicted in Fig. 5 (*Reinhold et al., 2024*). The *PDFUtilityFunction* provides an objective method for mapping scores across a wide range of distributions. Scores are calculated as follows. We first create a benchmark repository by assembling a collection of software artifacts and assessing each with one or more static analysis tools (Steps 1–3 in Fig. 5). Currently, the results from all static analysis tools for all software artifacts are stored in system memory, however, we are working toward storing these results in an aggregation file. This aggregation file will be generally preserved as a flat file wherein each row represents a software artifact in the benchmark repository and each column corresponds to a finding measured by a static analysis tool. The values in the cells will contain the results from the static analysis tool(s). We determine the density of scores for each finding in the benchmark by calculating a probability density function based on the distribution of the values in each column.

When an end user wishes to evaluate a new software artifact, they run it through the same SATs that the collection was evaluated with. Each finding is then scored by comparing it against the score against the density of scores in the benchmark repository. In the example in Fig. 5, the end user's software artifact of interest is found to have 40 instances of the
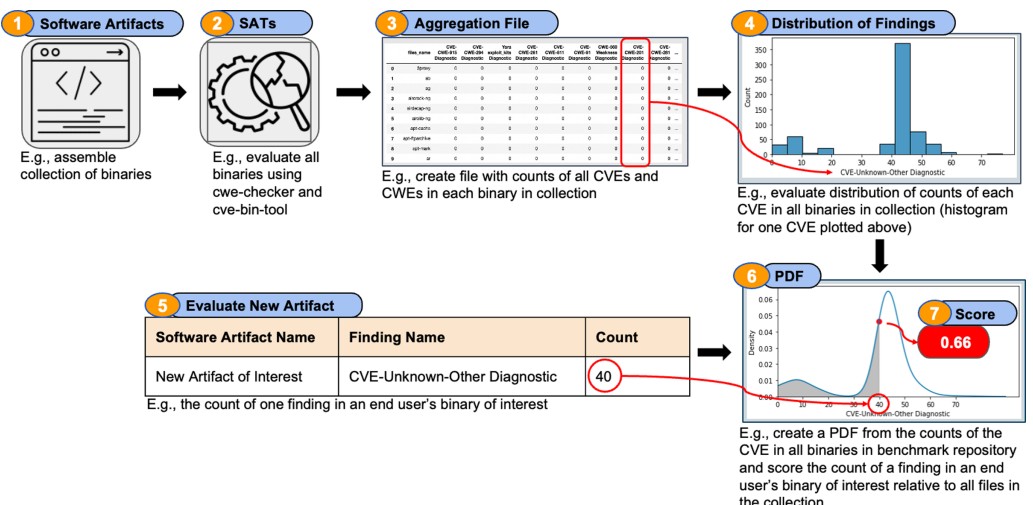

**Figure 5** **The density-based scoring procedure used in PIQUE Utility Function.** Step 1: software artifacts are assembled into a collection. Step 2: all software artifacts in the collection are evaluated using a cadre of SATs. Step 3: the findings for all software artifacts evaluated by all SATs are aggregated. Step 4: the distribution of the count of findings is created. Step 5: an end user runs their new artifact of interest through the same SATs as in Step 3 and the instances of each finding are recorded. Step 6: a probability density function (PDF) is created from the distribution in Step 4. Step 7: the count of the finding in the new software artifact of interest receives a score between [0,1] based on its position in the PDF. Figure and figure legend reproduced from *Reinhold et al. (2024)* and reproduced here from *Military Cyber Affairs* ©2024. Image source of step 1: https://www.flaticon.com/free-icons/source-file, and step 2: https://www.flaticon.com/free-icons/failure.

finding named "CVE-Unknown-Other-Diagnostic". The density-based scoring indicates that this artifact has fewer instances of this vulnerability than 66% of the artifacts in the collection.

The mathematical underpinnings of the PDFUtilityFunction are as follows. Each finding reported by SATs is treated independently when calculating the quality score. We use the "trapezoidal rule" by *Yeh (2002)* to approximate the definite integral representing the probability density function in Step 6 of Fig. 5 (Eq. (1)). The definite integral of a function represents the area under the curve between two specific points.

$$A = \int_a^b f(x)dx \simeq \frac{b-a}{2n}[f(a)+2f(x_1)+...+2f(x_{n-1})+f(b)] \tag{1}$$

We obtain $f(x)$ using kernel density estimator (KDE), a non-parametric method to estimate probability density function (PDF) from the counts of findings across all files in the collection (Steps 4–5 in Fig. 5). The PDF describes the probability of the count of the finding in the artifact of interest falling within the interval denoted by $a$ and $b$; $a$ is zero (as the minimum count of a finding in any artifact is zero) and $b$ is the upper bound (established by the software artifact in the collection having the highest count of a finding). By fundamental property, the total area under the curve of a PDF must be one.

Suppose an end user has a software artifact that they are evaluating and the count of a particular finding in that artifact is represented by $d$ (we call this $d$ because the count of a

finding is "diagnostic"; $d = 40$ in Fig. 5). We compute the area under the curve from zero to $d$; thus, $A_d$ indicates the proportion of samples in the collection having fewer counts of a finding than $d$.

The scaling method employed in PDFUtilityFunction is appealing because it is unbiased, repeatable, and effective at normalizing results from diverse sources (here, multiple SATs) to enable aggregation. However, it has some limitations. The external validity and accuracy of inference derived from this solution is heavily dependent on the benchmark repository— *i.e.*, the representativeness of the collection of artifacts against which a new artifact is being compared. In addition, the accuracy of the outputs of the PDFUtilityFunction is constrained by the accuracy of the SATs.

## Aggregation

A third component that can significantly affect quality scores is the variability associated with external sources to the PIQUE quality model. This variability can come from different tools and different versions of the same tools. Further, results produced by tools may come in different scales and different ranges.

The methods described in 'Normalization and Score Mapping' alleviate many of these challenges, however, as scores propagate up the hierarchy in a PIQUE model, a modeler is given an opportunity to weight scores from child nodes.

As discussed in 'Modern hierarchical models', the QATCH framework introduced the Analytical Hierarchy Process (AHP) by *Saaty (2008)* that uses pairwise comparisons between characteristics at the higher levels of a hierarchical model. We have found that using the AHP algorithm to weigh higher levels of a hierarchical model is a tedious process that stakeholders struggle to accomplish. Consequently, we are adopting the process of replacing the manual weighting with semi-automated approaches. Our current research led us to applying ML to algorithmically assign weights. This approach is showing great promise for learning the profiles of weight combinations based on documentation originating from benchmarks of tool findings and relevant standards. In 'Machine Learning' we provide an in-depth discussion of this approach. The result is that aggregated scores more accurately represent the quality of the target system.

## Machine learning
### Initial and default approach

Our initial approach to parameterizing the edge weights—and improving the real-world challenges faced by the AHP algorithm—featured an ordered weighted-average approach. In this approach, like the AHP, a stakeholder would create two sets of rank-based lists: one set reflecting the ranked importance of each product factor, and another indicating the importance of various quality aspects. Each item in each list captures the importance ranking of each item with respect to the item's parent node in the model. We applied numeric values to each item in each list based on the item's placement in the list normalized to the size of the list. This approach violates the operations that should be performed on ordinal scale data, yet our goal was to assess the validity of this approach for real-world solutions. We found that this method did not improve upon the challenges encountered by the AHP; specifically, ranking every product factor and quality aspect with respect to

importance was a challenging and tedious task for stakeholders. At present, this approach is implemented in all PIQUE models, and edge weightings of equal weights are supplied by default. However, we have begun experimenting with alternatives that will facilitate parameterizing the edge weights.

### Experimental approach

We are actively experimenting and applying ML methods to programmatically learn the edge weights between the measure, product factor, and quality aspect nodes in any PIQUE model. The goal of this ML approach is to assign appropriate weights that enable PIQUE to produce meaningful scores for each node, thereby enhancing the TQI and improving decision-making. Other approaches that experiment with improvements to decision making include Bayesian techniques such as work by *Manzano et al. (2018)*.

In our initial approach, the weights are assigned by manually analyzing the definition of each measure or product factor and creating a binary relationship with each quality aspect. The result of this approach is an indication of whether a given factor or measure impacts a quality aspect. Yet, the definition does not inform how much a measure affects a quality aspect relative to other measures and factors. Hence, the initial approach of assigning equal weights assumes that each measure/factor is independent, has an equal frequency of occurrence, and has an equal impact on software quality.

In the ML approach, we supplement the initial model with a benchmark dataset to establish an informed relationship between measure/factor and quality aspects. We look at the data distribution of the benchmark dataset and update the weights from the initial model to fit the distribution. This distribution provides an estimate of how the measures are implemented in practice. By fitting the initial model to the distribution, the final scores are relative to the benchmark repository, and better represent the project under analysis. Hence the scores have more meaning and can be compared to known software in the benchmark dataset. The ML approach can capture the dependencies between the measure/factors and identify trends in practical implementations. The modular design of PIQUE facilitates the generalization of this ML approach to all other PIQUE models, with ease.

As an example, consider the two software weakness measures CWE-117 and CWE-1024 in the PIQUE-bin model. Figure 6 shows the overview of the initial and ML approaches workflow. According to the definition, taken from the database of Mitre software weaknesses, these weaknesses affect the software security characteristic "Integrity". The definition only tells us that higher counts of CWE-117 (https://cwe.mitre.org/data/definitions/117.html) and CWE-1024 (https://cwe.mitre.org/data/definitions/1024.html) should be correlated with the "Integrity" aspect. Hence, the initial approach is to assign equal weights for the two measures and linearly increase the score with the number of findings. This model manifests as a 45-degree line in the orthogonal space of CWE-117 and CWE-1024. When a new project is under analysis, the findings of the SAT on the project are projected onto the line to calculate the score by normalizing. In contrast, the ML approach adjusts the initial model line using regression based on the findings of CWE-117 and CWE-1024 in the benchmark data (https://github.com/MSUSEL/benchmarks). This

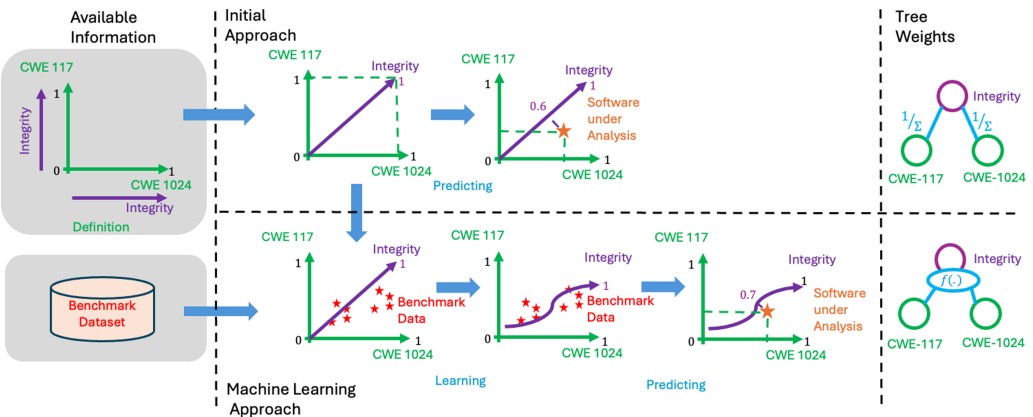

**Figure 6** **Workflow of Initial approach and Machine Learning approach, for the specific example of mapping CWE 117 and CWE 1024 to the quality aspect "Integrity".** Measures CWE-117 and CWE-1024 are drawn in the axes in an abstract space, and the quality aspect (Integrity) is represented as a function in that space. The left column shows the available data, the middle column shows the two approaches and the right column shows the PIQUE tree structure.

allows the "Integrity" characteristic to change based on the distribution of findings in the benchmark data, resulting in a more meaningful quality model.

Although we only considered two measures in the example above, the number of measures affecting a quality aspect is determined by the PIQUE model definition. Thus, the quality aspect "line" will be in a higher dimensional space consisting of all the measures contributing to a quality aspect. Further, since quality aspects are independent, separate models will be learned for each quality aspect (*e.g.*, "Integrity", "Authenticity", and "Availability"). Users can use different linear and non-linear regression ML models depending on the PIQUE model and the benchmark dataset. Due to the need for explainable ML models, classical low-order ML models are preferred over Deep Neural Networks. The ML models are intentionally trained to be over-fitted to the benchmark data, allowing the calculated score to be compared to an existing program in the benchmark. Consequently, the ML approach is sensitive to the benchmark data.

## USE CASES

We have used the PIQUE framework to develop several operationalizations, including models to assess the quality of C# (https://github.com/MSUSEL/msusel-pique-csharp), specialized security aspects of C# (https://github.com/MSUSEL/msusel-pique-csharp-sec), C programs (https://github.com/MSUSEL/msusel-pique-vendor), binary programs (https://github.com/MSUSEL/msusel-pique-bin), docker images (https://github.com/MSUSEL/msusel-pique-cloud-dockerfile), and SBOMs (https://github.com/MSUSEL/msusel-pique-sbom-supplychain-sec). Each model presents unique challenges, such as varying stakeholder requirements, tooling capabilities specific to the technology, and different deployment expectations (*e.g.*, in a Continuous Integration/Continuous Deployment (CI/CD) environment or as a standalone tool). This section illustrates two such models,

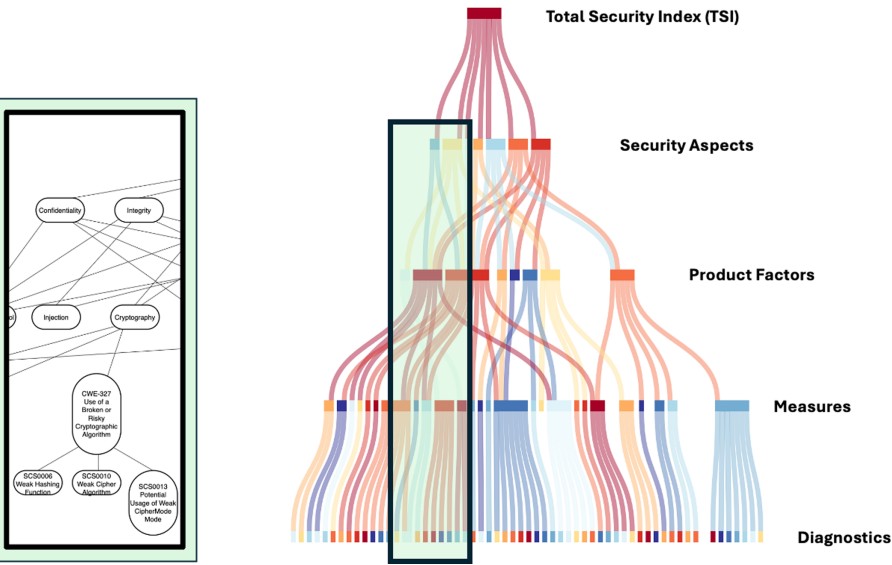

**Figure 7** **Partial view of the C# source code security model.** The right-hand side of the image displays the names of the various layers, spanning from the "Diagnostics" which represent tool outputs, to the Total Security Index (TSI) at the top of the tree that provides a holistic QA score. The complete security model contains 6 quality (security) aspects aligned with selected ISO/IEC 25010 standard (security sub-characteristics) and MS STRIDE, eight product factors, 25 measure, and 59 diagnostic nodes at its lowest level.

depicted in Figs. 7 and 8, which highlight their structural complexity. The layer naming conventions are based on *Wagner et al. (2012)*. In each illustration, we chose to focus on PIQUE models addressing only the code security quality aspect. Although our models can encompass all high-level quality characteristics (*e.g.*, ISO standards) and can be used in any domain, we intentionally highlighted code security. This decision was made for two reasons: First, security is an external quality characteristic that has received significant attention due to the increasing threats in digital technologies. Second, security is a complex characteristic that can be subdivided into additional layers of complexity with extensive support from Static Analysis Tools (SATs). The selection of security SATs varies across organizations. The PIQUE framework is designed to allow organizations to choose their own SATs tailored to their specific contexts, demonstrating the framework's independence and flexibility through these examples.

## Analysis of C# source code security

While many analysis tools exist that can be used to identify security vulnerabilities, the use of a quality model like the PIQUE model is beneficial in aggregating outputs from multiple analysis tools thus providing better coverage of security vulnerabilities (as compared to the use of a single tool). The aggregation of findings from multiple tools provides a broader security quality context accessible at multiple layers of the model. A partial view of the model is depicted in Fig. 7.

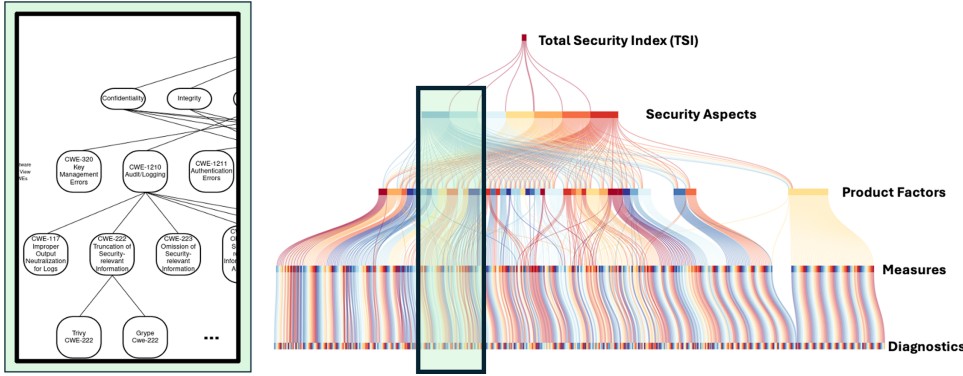

**Figure 8  Partial view of the SBOM security model.** The right-hand side of the image displays the names of the various layers, spanning from the "Diagnostics" which represent tool outputs, to the Total Security Index (TSI) at the top of the tree that provides a holistic QA score. The complete SBOM security model contains 7 quality (security) aspects aligned with selected ISO/IEC 25010 standard (security sub-characteristics) and MS STRIDE, 42 product factors, 403 measure, and 806 diagnostic nodes at its lowest level.

The requirements for the PIQUE security model were gathered by working with stakeholders directly. Their approval was obtained on design choices such as the static analysis tools used, the Security Aspects layer of the model, and the linkage between the Security Aspect layer and Product Factor layers. The model was created using a top-down approach, where the root note is the Total Security Index (TSI). This node decomposes into security aspect nodes, which are nodes taken from both the ISO/IEC 25010 standard *ISO (2011)* (security sub-characteristics) and the Microsoft STRIDE (https://learn.microsoft.com/en-us/azure/security/develop/threat-modeling-tool-threats) model.

The security model links findings from tools Security Code Scan and Insider. Although we also used the Roslynator tool during our exploratory phase, we decided to remove it because Roslynator diagnostics focused on code quality, while Security Code Scan and Insider focus on security-related findings. The Roslynator tool was only used during normalization to count the physical lines of code in the specified project.

We analyzed the performance of the security model by comparing its performance against 72 benchmark projects (26 open-source and 46 closed-source), with sizes ranging from 11 to 286,151 lines of code. We found significant evidence of a positive correlation between the size (lines of code) of a C# open-source project under analysis and its TSI. No similar conclusions could be drawn for closed-source projects.

We further validated the security model by investigating the effectiveness of the selected static analysis tools used within the model. We measured the ability of the selected tools to detect security vulnerabilities from the CWE Top 25 Most Dangerous Software Weaknesses by comparing all 59 Diagnostic nodes in our model to the CWE Top 25 list. Our security model has direct coverage for 52% of the CWE Top 25 Most Dangerous Software Weaknesses (2021 list). When we add the related responses to this count, coverage increases to 76% of the Top 25 list. Thus, the static analysis tools selected by the stakeholder were

effective when measuring and reporting vulnerabilities from the Top 25 list. An in-depth analysis of the vulnerabilities revealed that 7 diagnostics within the security model had the greatest impact on the TSI. A full description of this study can be found in *Harrison (2022)*.

## Analysis of software bill of materials security

An increasingly vulnerable software surface is the software supply chain (SSC). SBOMs enable the analysis of SSC security and is quickly becoming a fundamental cornerstone of SSC security. Assessing SSC security is achieved through scanning dependencies present in SBOMs and applying vulnerability mapping techniques to identify vulnerabilities present in SBOMs. As the SBOM space has evolved numerous SBOM analysis tools have been developed that assist in this process.

While these tools help identify vulnerabilities in SSCs, the use of a quality model like the PIQUE model is beneficial for multiple reasons. First as with other PIQUE models, aggregating outputs from multiple analysis tools provides broader coverage of security vulnerabilities through a holistic score. Additionally, SBOM presents a unique challenge when scanning for vulnerabilities. Due to the complex process of SBOM generation, we found that the process of generating an SBOM does impact the ability of SBOM analysis to find and report vulnerabilities (*O'Donoghue, Reinhold & Izurieta, 2024*), so using multiple SBOM analysis tools does help overcome this issue. Finally, each SBOM analysis tool is tailored to specific software ecosystems, potentially overlooking vulnerabilities in other ecosystems. Utilizing multiple SBOM analysis tools proves beneficial in tackling this challenge. Therefore, the aggregation of findings from multiple tools provides a broader security quality context that is accessible at multiple layers in the model.

The requirements for the PIQUE SBOM security model were gathered by working with stakeholders directly. Their approval was obtained on design choices such as the static analysis tools used, the Security Aspects layer of the model, and the linkage between the Security Aspect layer and Product Factor layers. The model was created using a top-down approach, where the root note is the Total Security Index (TSI). This node decomposes into Security Aspect nodes, which are nodes taken from both the ISO/IEC 25010 standard *ISO (2011)* (security sub-characteristics) and the Microsoft STRIDE model. These high-level Security Aspects are mapped to the category CWEs present in Mitre's CWE-699 Software Development view (https://cwe.mitre.org/data/definitions/699.html). As the CWE-699 view encompasses the software development lifecycle, organizing vulnerabilities in this way enables software providers to understand what aspects of their SSCs are weak in relation to software development. The model is large, and we provide a partial view in Fig. 8 that depicts some exemplary nodes at different levels. The visualization tool described in 'Visualization' can be used to view the entire model.

The security model links vulnerability findings from tools Trivy and Grype. We validated the security model by investigating the effectiveness of the selected static analysis tools used within the model. We leveraged mining software repository techniques to collect a large corpus of SBOMs (1,151) from common open-source repositories and Docker images with packages ranging from one to 4,135. Both Trivy and Grype reported a large number

of vulnerabilities across the SBOM corpus, however, vulnerability reports were rarely consistent between the tools as reported by *O'Donoghue, Reinhold & Izurieta (2024)*.

We are planning additional validation of the tools by gathering SBOMs containing 'ground truth' sets of vulnerabilities and comparing the vulnerability reports from Trivy and Grype against these established ground truths. Initial feedback from our collaborators has revealed several interesting findings, with one of the most interesting being the variations in model scores linked to the selection of SBOM generation tools and formats. These findings have spurred further discussion into the value that SBOMs provide.

## THREATS TO VALIDITY

There are several threats to the validity of the proposed study, which are grounded on the classification scheme of *Campbell & Cook (1979)*, *Campbell & Stanley (2015)*, and *Wohlin et al. (2012)*. We focus on (i) internal threats to validity, which refer to undesired relationships, and the extent to which independent variables cause effects on a dependent variable, (ii) external threats to validity, which describes the degree to which findings can be generalized (*i.e.,* statistically and ecologically), and (iii) construct threats to validity, which refer to how representative the study's measures, as captured by selected external tools, represent their intended real-world constructs (*i.e.,* meeting the representation condition of *Fenton & Pfleeger (1996)*).

Internal validity can only be mitigated through an interview process with experts in QA techniques for each respective case study. Although users of PIQUE models are primarily developers, they seldom have expertise in every characteristic used by the model. Although our use cases are validated by local developers, causal analysis remains a difficult problem. The visualization techniques provided with PIQUE do allow developers to investigate scores and weights in a model with more fidelity, however, complete mitigation of this threat is not possible.

External threats to validity are mitigated through the extensible PIQUE framework. PIQUE is designed to be agnostic of specific technologies thus allowing its operationalization in any domain, however, the external validity and accuracy of inference derived from PIQUE is heavily dependent on the representativeness of the collection of artifacts (*i.e.,* the benchmark) against which a new artifact is compared.

Finally, construct validity is dependent on the selection of tools that are representative of the theoretical characteristics that they intend to measure. The selection of tools is left to the domain experts.

## CONCLUSIONS AND SUMMARY

We aim to address the following research question:

### RQ

*Can we develop an extensible, flexible, and independent framework for operationalizing software quality across various domains, while enabling modelers to independently select their tools?*

Capturing the notion of quality is a wicked problem (*Rittel & Webber, 1973*) because of the inherent variability from multiple sources such as tools, tool versions, and measurement scales. Further, the subjectivity of a quality score is influenced by the mechanisms associated with the weighting of characteristics against each other, the lack of appropriate domain-specific benchmarks that can be used to calibrate scores, and the selection of tools that can act as proxies for theoretical characteristics (*e.g.*, ISO models). However, we posit that progress needs to be made in this space and PIQUE delivers in its ability to provide extensibility, flexibility and independence of operationalizations. Specifically, we deliver the following contributions with the PIQUE framework:

1. We extend the capabilities of prior art by specifying a new meta-model that allows for the operationalization of agnostic quality models that can be tailored to use any underlying standards such as ISO, STRIDE, CWE families.
2. We have developed improvements in aggregation techniques, benchmarking and utility function mapping. We have also improved on manual weighing approaches between layers of a hierarchical model by employing ML techniques that are trained on exemplary data.
3. We allow developers to integrate new tools, and leverage existing tools that can connect to PIQUE models.
4. We provide visualization technology that addresses concerns of high-level management as well as model developers. Visualization aids for the latter, to aid with debugging at individual node levels, are lacking.
5. We exemplify the deployment of PIQUE through two deployed use cases.
6. We provide an extensible metamodel of the PIQUE framework. The metamodel is designed with extensibility in mind.

PIQUE contributes to the academic community because it offers a platform that can be used to experiment with new techniques to assess the quality of systems. There are many potential areas for improvement including but not limited to aggregation techniques, weighing of characteristics, and visualization. The practitioner community also benefits because many organizations are unlikely to devote resources to developing these frameworks. They instead use "out of the box" technology that can be deployed quickly. Practitioners can either deploy out of the box or choose to dedicate additional resources to calibrate their models.

Finally, it is important to emphasize that future work in SQA is paramount given the proliferation of software technology. Everything is connected, from personal technologies to critical infrastructure, which creates a landscape where the ripple effects of a poorly constructed software component can propagate through the supply chain with unknown and untraceable consequences. For this reason, we must address software quality as a *first class citizen* and evangelize software quality *by design*. With PIQUE, we aim to continue to elevate the importance of software quality by incorporating newer techniques and technologies as they emerge. Significant improvements in machine learning and accessibility

to a larger open source corpora of examples are compelling reasons to continue to evolve the quality assurance landscape.

### Funding

This research was conducted with support from the U.S. Department of Homeland Security (DHS) Science and Technology Directorate (S&T) under contract 70RSAT22CB0000005. Any opinions contained herein are those of the author and do not necessarily reflect those of DHS S&T. This research is also supported by TechLink (TechLink PIA FA8650-23-3-9553). There was no additional external funding received for this study. The Department of Homeland Security Science and Technology (DHS S&T) division reviews our article to ensure no sensitive information is included, and to ensure that the scientific content is sound. This article contains unclassified work only, including all data sources and GitHub repos.

### Grant Disclosures

The following grant information was disclosed by the authors:
The U.S. Department of Homeland Security (DHS) Science and Technology Directorate (S&T): 70RSAT22CB0000005.
TechLink (TechLink PIA): FA8650-23-3-9553.

### Competing Interests

The authors declare there are no competing interests.

### Author Contributions

- Clemente Izurieta conceived and designed the experiments, performed the experiments, analyzed the data, prepared figures and/or tables, authored or reviewed drafts of the article, software design, and approved the final draft.
- Derek Reimanis conceived and designed the experiments, performed the experiments, analyzed the data, performed the computation work, prepared figures and/or tables, authored or reviewed drafts of the article, software and model development, and approved the final draft.
- Eric O'Donoghue conceived and designed the experiments, performed the experiments, analyzed the data, performed the computation work, prepared figures and/or tables, model developmnet, and approved the final draft.
- Kaveen Liyanage conceived and designed the experiments, performed the experiments, analyzed the data, performed the computation work, prepared figures and/or tables, authored or reviewed drafts of the article, aI and ML, and approved the final draft.
- A. Redempta Manzi Muneza analyzed the data, performed the computation work, authored or reviewed drafts of the article, aggregation technique algorithms, and approved the final draft.
- Bradley Whitaker conceived and designed the experiments, analyzed the data, prepared figures and/or tables, aI and ML, and approved the final draft.

- Ann Marie Reinhold conceived and designed the experiments, performed the experiments, analyzed the data, prepared figures and/or tables, authored or reviewed drafts of the article, data science, tool evaluation, aggregation algorithms, and approved the final draft.

## Data Availability

All source code is available at GitHub: https://github.com/MSUSEL/msusel-pique.

The Machine Learning data is available at GitHub: https://github.com/MSUSEL/benchmarks

All instantiations of models and use cases, also listed as footnotes, are available at GitHub: https://github.com/MSUSEL.

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
