# Peer review of "A Generalized approach to the operationalization of Software Quality Models"

_PeerJ Computer Science, doi:10.7717/peerj-cs.2357_

## Round 0.1 · original submission · Minor Revisions

The reviewers agreed that the paper is clear and that the topic is very relevant. The paper is technically solid, but the presentation must be improved before publication, and some related work added. In addition, some parts of the paper must be clarified. The amount of work to be done amounts to a minor revision, carefully addressing the detailed comments provided by the reviewers (in particular those of Reviewer 2).

·

Basic reporting

The paper proposes a novel approach that includes a hierarchical software quality assurance (HSQA) model, aiming to enhance the calibration, validation, and operationalization of quality models. The method incorporates advanced aggregation techniques, machine learning for weight selection, and improved tool selection processes. Furthermore, the study introduces a developer-focused visualization tool to aid in understanding and debugging quality models, providing a clear visualization of how different choices affect the model at various levels. The manuscript is well-written, and the premise of the study is clearly conveyed

Experimental design

The PIQUE model structure and approach are comprehensively detailed and well-articulated. However, the evaluation section (i.e., section 4) could benefit from greater clarity. It would be helpful to provide a rationale for focusing the evaluation specifically on “Code Security.” Explaining the choice to limit the scope to this particular aspect could enhance the reader's understanding of the evaluation criteria and the relevance of “Code Security” within the broader objectives of the PIQUE model.

In addition, it would be beneficial to explicitly state the research question (RQ). A clearly defined RQ is essential in guiding readers and researchers through the narrative and focus of the study. Presenting a well-articulated RQ would not only align with the traditional expectations of empirical investigations but also enhance the ability to assess the effectiveness and thoroughness of the study's findings. Clarifying this could significantly strengthen the paper's framework and help readers evaluate how well the study addresses its core objectives.

Validity of the findings

Evaluating these types of models within the confines of a review process presents certain challenges. However, after reviewing the provided GitHub repository, it appears that additional guidance on how to run the model could enhance its accessibility and usability. Detailed instructions or a demonstration using a specific use case could greatly aid in operationalizing the model effectively. Apart from this, the combination of empirical evidence and detailed description in the paper does meet the reporting rigor required for studies of this nature.

Additional comments

The authors aptly describe measuring “quality'” as a “wicked” problem, a sentiment echoed in my own research experiences, which suggest that the relationship between “software quality” and product engineering is not always linear. Historically, the impact of quality models and metrics has often been underwhelming. In light of this, a more profound reflection on the effectiveness and necessity of developing new models would be beneficial. A discussion about the outcomes and the motivations for continuing to develop these models could elevate the manuscript depth. One potential criticism is that this is "just another model"! Regardless, the study demonstrates significant effort and creativity.

Reviewer 2 ·

Basic reporting

In general, the paper is well-organised and it is easy to read. But I miss a kind of continuity, i.e., the text is written as separate short sentences, I miss some “flow”. There are also some “generic” sentences that I’m sure that I was not able to understand what the authors wanted to express, for example:
- line 43 “…to ensure that the tool does not threaten the construct validity of the theoretical quality attribute the tool is intended to measure.”
- line 60 “…Data transformations can threaten the internal and conclusion validity of an SQM because aggregation involves assessing all measurements up the hierarchy using the edge weights provided by the model.”
- line 228: “Numerous tools inform HSQA models.”

Authors start talking about tools to measure before mentioning any example of what kind of property they want to measure. It would help to the reader have some example of quality attribute to be measure and the kind of metric needs to be used to better understand the kind of tools the authors are referring in the first pages of the paper. The first example is included in line 57, with no reference to the software quality attribute that can be measured. I understood the introduction, but I believe that it is because I am an expert on quality models and acquainted on using data from several tools (git, project management tools, …) to feed metrics to measure quality attributes, it can be difficult for non-experts. The use of concepts like aggregation (line 57) or normalised measures (line 58) without concrete examples are difficult to understand.

Regarding the organisation, in section 2, a brief generic view of the full approach would help the readers to understand the role played by the different components described in this section. In section 2.1 (line 288) introduced a 3-step PIQUE quality assessment “(1) Deriving a quality model, (2) Benchmarking the quality model based on similar projects, and (3) Using that model to assess the quality of a system.”. Maybe introducing this 3-step “process” and the relation with the other artefacts (e.g., is this quality derivation related to the 3-step process shown in figure 2 and section 2.3 subsections) would be helpful. The PIQUE framework metamodel (Figure 3) is introduced to early, it cannot be understood without the software components explanation (section 2.1.2). Maybe a figure with the software components could also help to understand the tool arquitecture.

The literature references are correct, I only miss some references (of at least that authors would review the papers related to the ones I include in my general comments section). I agree on the claim “Capturing the notion of quality is a wicked problem (Rittel and Webber (1973))” (line 678), but I would call on the authors to look for a newer reference. The brief history of software quality modelling provided in section 1 provides a good view of how the topic of this paper is large addressed by researchers is not already solved.

Regarding the figures, I miss a figure depicting the PIQUE generic process (involving all the steps and artefacts) and the PIQUE tool architecture.

Minor comments:
- In the first page, authors introduce the acronym SQM (Software Quality Model), but from page two on, they use SQA models. At the end of page two, authors mentioned HSQA for Hierarchical Software Quality Assurance, so maybe this SQA is Software Quality Assurance.
- TQI acronym is not introduced. I found the meaning in Figure 4 (Total Quality Index).

Experimental design

A short section explaining the research method would be appreciated.

Validity of the findings

no comment

Additional comments

The topic of operationalising the quality assessment is an interesting topic for the community. Software quality “definition” is widely addressed with several proposals (and standards), but there is a lack on the operationalisation on concrete metrics and tools to automate this quality assessment. The PIQUE approach is promising.

I have some doubts about the distinction between SQM and HSQA in the introduction, the software quality models (SQM) like the ISO/IEC 25010 are already “hierarchical”, the standard defines the software quality as a 2-level model defining characteristics and sub-characteristics, metric measuring the sub-characteristics would be the third level. The Quamoco approach mentioned by the authors defines a 3-level quality model (quality, factor, measure).

I’m not sure if I agree to the authors when they claim that “The comparison and contrast of these tools is beyond the scope of this paper; …” (lines 228-229, section 1.2). If there are available tools that produce metrics to measure concrete software quality, it seems quite related to the topic of this paper (those software tools are operationalising the measurement of concrete quality attributes). In fact, as authors already mention, Sonarqube uses SQALE quality model for assessing technical debt.

I understand (and agree on) the PIQUE 3-level model, but I cannot appreciate too much the difference with the ISO/IEC 25010 (characteristics, sub-characteristic and metrics). In fact, I do not find too much difference to the quality model proposed by [1] in 2019 with Strategic Indicators (Security Score in Figure 1), Factors (High-level business goals in Figure 1), Metrics & raw data (measurable concepts in figure 1), and data source (tool output in figure 1). It is true that [1] does not include ML algorithms for calculating weights, but the related work [2] is using Bayesian networks for this issue. The only aspect that I could argue is naming high-level “business” goals to concrete security aspects.

I have some concerns with the activity of “model benchmarking”, I understand benchmarking (in this context) as a step to measure the model efficiency or performance. So, you run the model, and see if the resulting assessment corresponds to the expected assessment. But, it seems that this is not what this step is about. The authors explain that the resulting models from the benchmarking can be used as input for the model assessment. After reading section 3.4 I understand (If I understand well) that this benchmarking is for getting the edge weights. I believe that if the “general process” is presented previously (like I mentioned in my Basic reporting comments related to the organisation) all the steps will be better understood.

It is quite “bold/reckless” claiming that “Notably lacking in quality assurance, is the existence of visualization tools” (line 415). It is true that visualisation issues need some improvement in this field, but there are plenty of software analytics tools including this kind of quality visualisation. Concretely, related to “…is a lack of similar counterparts available with frameworks”, related to [1] there is an associated tool providing a quality assessment dashboard.

One of the authors benefits claimed by the authors is already claimed by [1] “the use of a quality model like the PIQUE model is beneficial in aggregating outputs from multiple analysis tools thus providing better coverage of security vulnerabilities” (lines 586-587).

The use cases presented by the authors cover the expectations, maybe presenting the use cases at the beginning and using some of the aspects, product factors, measures, and diagnostics when the approach modules are presented would improve the understandability.

[1] Continuously Assessing and Improving Software Quality with Software Analytics Tools: A Case Study. IEEE Access 7: 68219-68239 (2019)
[2] Using Bayesian Networks to estimate Strategic Indicators in the context of Rapid Software Development.

Cite this review as

---

## Round 0.2 · accepted · Accept

I carefully read the revision and the authors' response to reviewers. The authors addressed all reviewers' comments and improved the overall presentation of the paper. The paper is ready for publication.